# Frankenstein Optimizer: Harnessing the Potential by Revisiting Optimization Tricks

## Abstract

Gradient-based optimization drives the unprecedented performance of modern deep neural network models across diverse applications. Adaptive algorithms have accelerated neural network training due to their rapid convergence rates; however, they struggle to find optimal generalization compared to stochastic gradient descent (SGD). Many prevailing adaptive optimizers, though often building upon Adam for enhancements, typically have their momentums statically defined at the beginning of training. Thereafter, they offer little to no capability for effective dynamic adjustment in response to the highly dynamic training conditions. By revisiting various adaptive algorithms' mechanisms, we propose the Frankenstein optimizer, which combines their advantages. The proposed Frankenstein dynamically adjusts first- and second-momentum coefficients according to the optimizer's current state to directly maintain consistent learning dynamics and immediately reflect sudden gradient changes. Extensive experiments across several research domains such as computer vision, natural language processing, few-shot learning, and scientific simulations show that Frankenstein surpasses existing adaptive algorithms and SGD empirically regarding convergence speed and generalization performance. Furthermore, this research deepens our understanding of adaptive algorithms through centered kernel alignment analysis and loss landscape visualization during the learning process.

## 1 Introduction

In recent years, neural networks (NNs) have become increasingly prevalent in various fields, driving the widespread adoption of gradient-based optimization methods in science and engineering. This increased usage has led to studies enhancing optimizers to achieve faster convergence and better results with limited computational resources. Among these optimizers, AdamKingma (23) has emerged as the most prominent, often serving as the default setting in deep learning tasks. However, many studies have highlighted that Adam-like optimizers do not necessarily achieve superior generalization(70; 34), with simple experiments(60) demonstrating this issue. As a result, the debate between Adam and stochastic gradient descent (SGD) has persisted within the deep learning community.

The rapid growth in the parameter size of deep learning models has magnified the importance of adaptive optimizers despite their known limitations (55). Current state-of-the-art large language models (LLMs), such as Nemotron-4 with 340 billion parameters (2), LLaMA 3.1 with 405 billion parameters(13) and DeepSeek-V3 with 671 billion parameters (30), exemplify the growing scale of modern architectures. Considering the computational cost for training GPT-3 with 175 billion parameters (5) which amounts to $4.6 million, it is obvious that the efficiency of optimization algorithms directly impacts AI economics.

Adam or its variants have been actively developed to design faster optimizers. For example, AdaBelief(71) demonstrates its superiority in addressing deep valley problems by simultaneously considering both adaptive terms and momentum. Similarly, Sophia optimizer(31) showcases its ability to manage risks in the update

process by correcting the Hessian, ensuring its stability, and yielding promising results in large models like LLMs. However, the adaptive nature during optimization remains underexplored, leaving room for further improvement. To further explore these adaptive adjustments and the potential for improvement, we propose a new optimizer, "Frankenstein," developed on the techniques used by previous gradient-based adaptive optimizers, while incorporating our three key innovations:

**1) Adaptive momentum coefficients** $\beta$: Unlike previous optimizers that employ a fixed momentum coefficient, our approach dynamically adjusts the momentum based on the current state and learning rate schedule. This design enhances the adaptability of the optimization process, enabling more efficient convergence across diverse training scenarios.

**2) Relaxation strategy for second momentum** $v_t$ : Along with adjusting the second momentum, we propose a novel relaxation strategy that balances the maximum recorded value, as utilized in AMSGrad variants (41), with conventional unconstrained exponential moving average methods. This approach accelerates optimization while mitigating convergence risks, resulting in significant improvements in solving adaptive problems.

**3) Acceleration factor** $\xi$: For fast and accurate convergence, we designed an acceleration factor that comprehensively evaluates the system's gradient magnitude and degree of belief, as well as its proximity to convergence. It determines how much the gradient is reflected in the parameter update.

To illustrate the impact of our design principles, we plot the optimization trajectories of several established adaptive algorithms and Frankenstein on a representative loss surface (Fig. 1). Under large-gradient conditions, Frankenstein accelerates learning by up to $1.6\times$ the base rate. As convergence nears, updates become dominated by the rapidly decaying second momentum term, shifting the emphasis to the nonlinear misalignment factor $P$. This transition produces two distinct peaks in the normalized adaptive coefficient, defined as the multiplier applied to momentum or gradients during updates to modulate the effective learning rate. Tracking this coefficient reveals the optimizer's dynamic preferences. Moreover, this switching mechanism ensures stable descent across flat or plateaued regions, a common challenge in quantum neural network training(38). Finally, as gradients vanish and $P$ stabilizes, the coefficient $\xi$ returns to one, restoring the non-accelerated update regime.

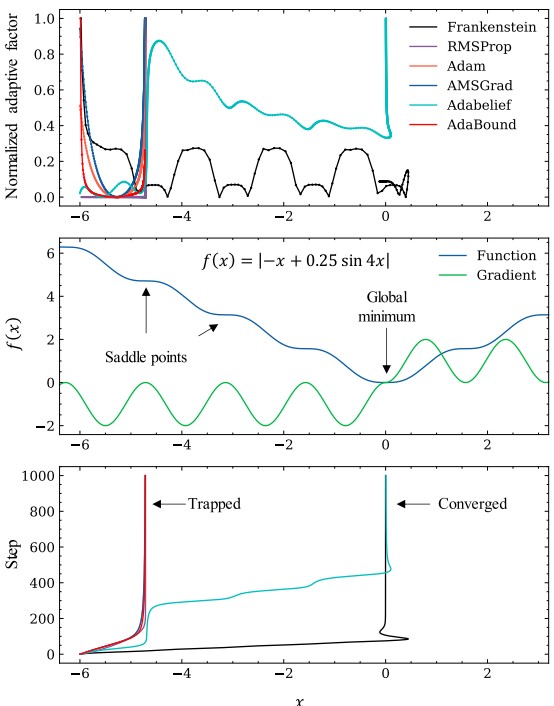

## 2 ALGORITHM

### 2.1 ADAPTIVE
#### FIRST MOMENTUM COEFFICIENT $\beta$

Adjusting the learning rate throughout NN training is a common strategy to improve model performance. Research indicates(45; 63; 17) that there should be a consistent "noise scale" across batch

Figure 1: From top to bottom: (1) Adaptive factor for each optimizer during the optimization process of $f(x)$; (2) Function $f(x)$ and its corresponding gradient; (3) Historical trajectories of each optimizer as a function of optimization steps.

size, learning rate, and momentum. Therefore, our proposed algorithm also adapts the momentum coefficient based on the current learning rate. In Verlet integration, $\Delta x$ is affected by the current momentum, gradient, and learning rate, as illustrated in Equation 1. Let $\alpha$ represent the learning rate, $m_t$ represent the first-order momentum, $g_t$ represent the gradient, and $\theta$ represent the learning parameter.

$$\theta_{t+1} = \theta_t + m_t\alpha + \frac{1}{2}g_t\alpha^2 \tag{1}$$

If the training process enters a stable phase where the gradient remains relatively constant and the momentum closely matches the gradient, the momentum coefficient should be maintained as specified in Equation 2; otherwise, this balance could be disrupted.

$$(1-\beta)m_t \approx \frac{1}{2}g_t\alpha^2 \tag{2}$$

To maintain training consistency, the decay rate caused by momentum and the momentum coefficient should correspond to the gradient and the learning rate; otherwise, using a learning rate decay strategy might lead to overly rapid convergence, requiring additional iterations to achieve the optimal convergence point. Based on this insight, we propose a momentum coefficient adjustment strategy, detailed in Equation 3. This formula reflects common configurations, where training from scratch uses $\alpha = 10^{-3}$ and $\beta = 0.9$, while fine-tuning adopts $\alpha = 2 \times 10^{-4}$ and $\beta = 0.9$. In our algorithm, each time the learning rate changes, a new optimal $\beta^*$ is calculated based on the corresponding learning rate instead of using a fixed $\beta$.

$$\beta^* = 1 - (1-\beta)\sqrt{\frac{\alpha}{10^{-3}}} \tag{3}$$

It's worth noting that other algorithms update momentum by applying a fixed-coefficient moving average, which retains information from multiple previous steps during iterations and smooths the trajectory. This approach takes several steps to approximate the ideal state. In contrast, our Frankenstein can directly maintain consistent learning dynamics. In other words, it isn't forced to converge due to changes in the learning rate; instead, it decides whether to converge based on the actual gradient conditions (with the final momentum coefficient still remaining $< 1$).

Here, we define the nonlinear misalignment factor ($P$), which indicates a misalignment between the past momentum $m_{t-1}$ and current gradient $g_t$. $P$ takes a value between 0 and 1, and it becomes smaller when the two vectors are well-aligned in the same direction, while it becomes larger when they are aligned in opposite directions. $P \leftarrow \cos^{-1}(\tanh(m_{t-1} \odot g_t))/\pi$. Additionally, the we propose another parameter $\rho$ to adjust the momentum coefficient. $\rho \leftarrow \log(e^1 + \sqrt{v_t} + 0.5 - P)$ where $x_t$ refers the summation of squared gradient $g_t^2$ and a small constant $\epsilon$. The $\rho$ parameter controls whether the momentum coefficient should increase based on the difference between the current system state and the ideal convergence state. The coefficient is influenced by two conditions: (1) whether the squared gradient approaches zero and (2) whether the $P$ factor approaches 0.5. As these conditions near the convergence state, the momentum coefficient will gradually decrease to the default value of $\beta$, effectively rendering it inactive.

Notably, the momentum coefficient can be adjusted to zero if the product of the momentum direction and the gradient is highly negative. This mechanism is inspired by the FIRE algorithm(18), where a large directional discrepancy may indicate that the step size is too large, causing a lack of convergence. Therefore, this feature includes reinitializing the momentum or reversing its direction when necessary.

## 2.2 SECOND MOMENTUM $v_t$ UPDATE WITH DYNAMIC EMA

The update rules for second momentum $v_t$ enable various strategies to achieve optimizers' adaptivity. The general process is outlined in the formula below.

$$g_t \leftarrow \nabla_\theta f_t(\theta_{t-1}) \tag{4}$$
$$v_t \leftarrow V(v_{t-1}, g_t, \beta, ...) \tag{5}$$
$$\theta_{t+1} \leftarrow U(\theta_t, v_t, \alpha, ...) \tag{6}$$

where $V$ represents a momentum update function and $U$ represents a parameter update function.

Starting with the basic RMSProp strategy, which applies a simple Exponential Moving Average (EMA) to the squared gradients, several adaptations have been developed (Appendix G). These methods depends on a fixed $\beta_2$ parameter throughout the training process. In contrast, this work proposes a strategy that also relies on squared gradients as its foundation but applies an EMA coefficient that adapts based on both current and historical states. This coefficient is determined by whether the $P$ factor approaches 0.5 and the ratio of the current to past squared gradients. The following paragraphs explain the motivations for this adjustment.

These Adam-like optimizers employ adaptive methods with second-order momentum, using a higher momentum coefficient for EMA compared to first-order momentum. This setup not only facilitates the transition of the Adam phase from divergence to convergence(69; 41) but also allows a larger $\beta_2$ to enhance long-term memory, smoothing the training process and helping to prevent issues like gradient instability. However, as NN parameters scale up, LLMs now use a $\beta_2$ value of 0.95 instead of the previous 0.999.

Even when addressing instability in the training process of large models by tuning $\beta_2$, potential spikes in the training trajectory may still occur, where a fixed $\beta_2$ can cause the adaptive ratio, $r_t = \frac{m_t}{\sqrt{v_t}}$ to shift from a unimodal distribution approaching zero gradient to a bimodal one, which may destabilize LLM training. In other words, fixing $\beta_2$ establishes a range within which gradient fluctuations remain manageable, especially avoiding sharp increases. Otherwise, a slow EMA could introduce bias in the adaptive coefficient, ultimately harming model performance.(39; 49)

Our dynamic $\beta_2$ design addresses challenges with the coefficient by leveraging the current-to-past ratio of squared gradients. This mechanism enhances adaptability and strengthens convergence stability by allowing immediate reflection of sudden gradients in $v_t$.

$$\beta_2 \leftarrow 1 - \frac{x_t}{x_{t-1}}|0.5 - P| \tag{7}$$

Notably, in cases where the current-to-past squared gradient ratio exceeds 2, $\beta_2$ may even take on negative values, further enhancing adaptability during drastic gradient changes. This mechanism outperforms a simple $\beta_2$ setting of zero by providing better adaptive control, as shown in the equation below.

$$v_t = g_t^2 - \beta_2(g_t^2 - v_{t-1}), \quad \text{where } \beta_2(g_t^2 - v_{t-1}) > 0 \tag{8}$$

Table 1: Summary of conditions: behavior of $\rho$ based on gradient square $x_t$, factor $P$ and $\beta_2$.

| Condition | $x_t$ | $P$ | $\rho\beta_1$ | $\beta_2$ | Optimizer Implication |
|---|---|---|---|---|---|
| Early Training | Large | $\sim 0$ | Large ($\sim 0.95$) | $\sim 0.5$ | SGD with high momentum |
| Mid-Training | Moderate | Increasing | Decreasing | Increasing | Adam with low $\beta_2$ |
| Near Convergence | Small | $\sim 0.5$ | $\sim 0.9$ | high $\beta_2$ | Adam with high $\beta_2$ |
| Gradient Spike | Very large | $> 0.5$ | $< 0.9$ | very low $\beta_2$ | SGD with low momentum |
| Vanishing Gradients | Very Small | $\approx 0.5$ | $\sim 0.9$ | $\approx 1$ | Adam with ultra high $\beta_2$ |

## 2.3 ACCELERATION FACTOR $\xi$

The adaptive coefficient has generally been influenced by both first- and second-order moments. However, alternative methods have been proposed to address challenges such as the risk of convergence to local optima and generalization issues. For instance, the Tiger(47) and Lion(8) optimizers rely on the sign of the gradient for adaptation, while Adabelief(71) replaces the squared gradient with the difference between the gradient and momentum. In this study, we tackle similar issues by introducing an additional parameter(14), $\xi$, which is estimated using the following equation.

$$\xi \leftarrow \frac{1 + e^{-0.5}}{1 + e^{-|x_{t-1} - P|}} \tag{9}$$

The parameter $\xi$ is computed based on the final difference between the squared gradient's magnitude and the $P$ factor. This estimation helps determine whether convergence has been reached and whether to adjust the learning rate accordingly. In the example shown in Fig.1, the gradients exhibit periodic and dramatic changes, which often cause most optimizers to get trapped in a local optimum due to rapidly diminishing gradient values.

However, the term $|x_{t-1} - P|$ enables the process to be dominated by $x_{t-1}$ under normal conditions, while the $P$ factor takes precedence during rapid gradient changes. This mechanism reduces the risk of converging to a local minimum. As depicted in the figure, the Frankenstein adaptive factor demonstrates a recurring pattern, with the $P$ factor dominating $\xi$ during an "idling phase" until a significant gradient change occurs.

**Numerical Stabilization** Clipping $(1 - \beta)$ to $[0.05, 0.99]$ gives $\beta_{1,t} \in [0.01, 0.95]$; with $\rho_t \in [0.8, 1.05]$ (from log–clip $[e^{0.8}, e^{1.05}]$) we ensure $\rho_t \beta_{1,t} < 1$, so momentum is contractive and acceleration bounded across LR changes. For the second moment, $v_t = \beta_{2,t} v_t + (1 - \beta_{2,t}) x_t$ is a first–order IIR; allowing small negative $\beta_{2,t}$ with $|\beta_{2,t}| < 1$ is stable and speeds re-centering after spikes. Thus, the term is non-decreasing—an AMSGrad-like safeguard under extreme transients.

## 3 EXPERIMENTAL RESULTS

### 3.1 IMAGE CLASSIFICATION

**Learning from scratch** We assessed the performance using the ImageNet-1K(44) dataset. With the timm(59) framework, we trained both ResNet18 and ResNet50(19) models, repeating the training process five times. Given that many studies evaluating optimizers compare ImageNet training over 200 epochs, it is impressive that Frankenstein achieves comparable results to Adam in just 40 epochs, as show in Table 2. It is also comparable to state-of-the-art optimizers like Adan and AdaFisher in just 80 epochs in ResNet18 model architecture.

**Fine-tuning** Transfer learning across different domains has become a widely adopted approach in deep learning. In this study,

---

**Algorithm 1** Frankenstein Optimizer

**Require:** Learning rate $\alpha$
  **Initialize** $\theta_0, m_0, v_0, t \leftarrow 0, x_0 \leftarrow \alpha, \alpha_0 \leftarrow \alpha$
  **repeat**
    $t \leftarrow t + 1$
    $g_t \leftarrow \nabla_\theta f_t(\theta_{t-1})$
    $\beta_{1,t} \leftarrow 1 - \text{Clip}\big(0.1\sqrt{\frac{\alpha_t}{10^{-3}}}, 0.05, 0.99\big)$
    $P \leftarrow \frac{\cos^{-1}(\tanh(m_{t-1} \odot g_t))}{\pi}$
    $x_t \leftarrow g_t^2 + \epsilon$
    $v_t \leftarrow \max(v_{t-1}, x_t)$
    $\rho \leftarrow \log\big(\text{Clip}(e^1 + \sqrt{v_t} + 0.5 - P, e^{0.8}, e^{1.05})\big)$
    $\xi \leftarrow \frac{1 + e^{-0.5}}{1 + e^{-|x_{t-1} - P|}}$
    $m_t \leftarrow \rho\,\beta_{1,t}\,m_{t-1} - \frac{\alpha_t\,g_t\,\xi}{\sqrt{v_t}}$
    $\theta_t \leftarrow \theta_{t-1} + \beta_{1,t}\,m_t - \frac{\alpha_t\,g_t\,\xi}{\sqrt{v_t}}$
    $\beta_{2,t} \leftarrow 1 - \frac{x_t}{x_{t-1}}\big|0.5 - P\big|$
    $v_t \leftarrow \beta_{2,t}\,v_t + (1 - \beta_{2,t})\,x_t$
  **until** $\theta_t$ converged

---

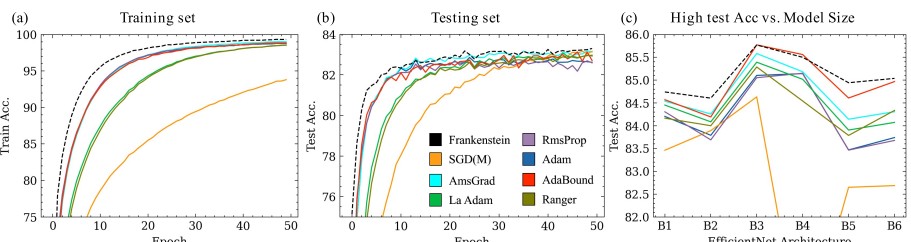

Figure 2: Testing set accuracy profiles of various optimizers on EfficientNetB0 with (a) CIFAR-10 and (b) CIFAR-100. (c) Testing set accuracy according to architectures (EfficientNetB0-6).

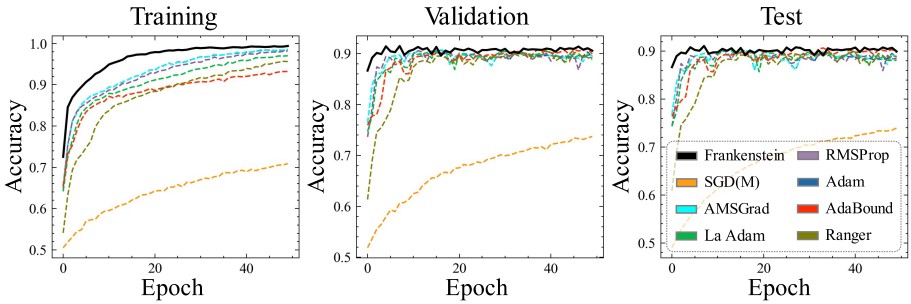

Figure 3: Training, validation and testing set accuracy with different optimizers dealing with IMDB movie reviews using BERT.

we trained various optimizers on Efficient-Net architectures ranging from B0 to B6, fine-tuning parameters originally learned from ImageNet-1K on the CIFAR-100 dataset. We reconfigured the initial classifier with a MLP layer to align with the target number of classes. Notably, we applied full parameter fine-tuning throughout to achieve optimal performance. Frankenstein got the best accuracy for B1, B2, B5, and B6 while providing the fastest convergence as show in Fig.2.

## 3.2 LANGUAGE MODELING

**Natural language understanding**  We fine-tuned a BERT model(12) on the IMDB movie review dataset(35) for binary sentiment classification. Two gated recurrent unit (GRU) layers(10) were appended to the transformer backbone(54) to produce the final prediction.

The data were split into training (70%), validation (30%) and a held-out test set reserved exclusively for evaluation. All experiments used the optimal learning rate, batch size of 128, and 50 training epochs. To eliminate random variability in the testing process, we conducted five experimental runs and averaged the results. Figure 3 shows that Frankenstein converged

Table 2: Validation Top-1 error rate on ImageNet-1K classification. *(71), +(7), ⊕(64). The standard deviation across multiple tests is presented in the benchmark.

| Optimizer | ResNet18 | ResNet50 |
|---|---|---|
| Adam (23) | 33.46* | 23.10⊕ |
| Lookahead-Adam (67) | – | 24.51 |
| Radam (32) | 32.38* | - |
| AMSGrad (41) | 32.31+ | - |
| Adabound (33) | 31.87* | - |
| PAdam (7) | 29.93+ | - |
| AdaBelief (71) | 29.92* | - |
| SGD (43) | 29.77* | 23.00⊕ |
| LAMB (66) | 31.36⊕ | 23.00⊕ |
| AdaFisher (37) | - | 22.99 |
| Adan (64) | 29.10⊕ | **21.90**⊕ |
| Frankenstein-40 epoch (Ours) | 31.24 ± 0.07 | 25.49 ± 0.11 |
| Frankenstein-80 epoch (Ours) | 29.13 ± 0.13 | 22.90 ± 0.05 |
| Frankenstein-120 epoch (Ours) | **28.75** ± 0.10 | 22.44 ± 0.06 |

most rapidly and delivered the strongest generalization performance among the compared models.

**Natural language generation** The instruction fine-tuning of the LLaMA-7B(53) using Parameter-Efficient Fine-Tuning (PEFT)(36) will serve as a benchmark test. LoRA(22) is integrated by adding branches with rank dimension 8 to each layer's `q_proj` and `v_proj` modules. The Alpaca dataset(50), generated via self-instruct(57) from `text-davinci-003`, used while 2,000 samples reserved for the test. The model is trained with a batch size of 120 for 3 epochs with a cosine annealing schedule, and the performance is evaluated using maximum token lengths of 256 and 512. As shown in Fig.4, Sophia achieves lower training loss than Frankenstein but worse test loss, indicating overfitting. In addition, the Adam optimizer is affected by grouping sequences based on their length during training, which results in performance spikes on the test set. This behavior reflects an underlying issue of training instability.

### 3.3 MATERIAL MODELING AND SIMULATIONS

Gradient-based optimization plays a crucial role in scientific simulations as well as deep learning, closely linked to simulation accuracy.

**Energy minimization** In energy minimization experiments, it is necessary to minimize both the atomic position to reduce corresponding potential energy and ensure that the model is reasonable at the start of the simulation. This process involves multiple iterations until the energy converges. The classic Lennard-Jones potential problem with 38 atoms is demonstrated during the study. The experiment involves randomly initializing 1,000 structures and testing for convergence when the total force drops to the

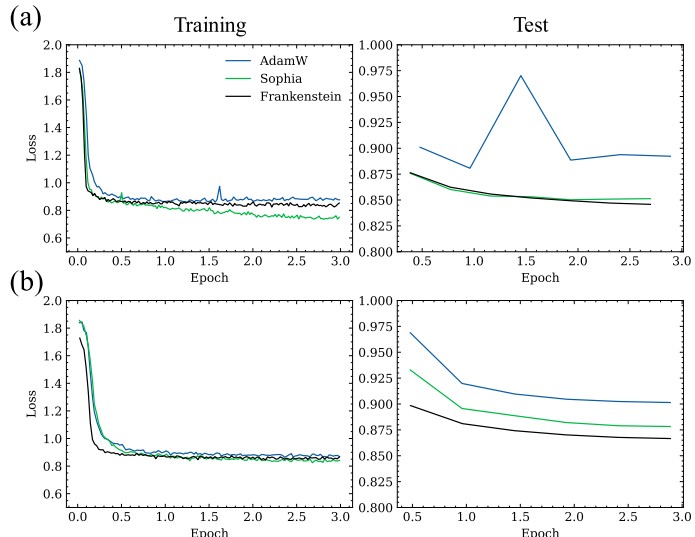

Figure 4: Training and testing losses for text lengths (a) 256 and (b) 512 from the instruction-tuning process of the LLaMA-7B model.

$1 \times 10^{-2}$ level. By analyzing the average, minimum, and maximum number of steps required by common algorithms, the differences between algorithms in molecular dynamics during energy minimization are highlighted. The entire energy minimization process is performed with Atomic Simulation Environment (ASE)(26) framework, with all initial atomic structures sourced from Lennard-Jones 38 of OptBench(6). In this benchmark, the number of force calls needed to reach the global minimum is recorded. Frankenstein reaches the global minimum with the least average steps as shown in Table 3. However, the maximum step count is relatively high, possibly due to the different structure of the energy landscape compared to that with flat minima in deep learning.

To investigate the case of more complex structures, a polycrystalline high-entropy alloy (HEA) model composed of Co, Ni, Cr, Fe, and Mn is generated using Atomsk(20). The energy minimization of the entire system uses the Second Nearest-Neighbor Modified Embedded-Atom Method (2NN MEAM)(27; 21) potential. The system consists of 5.5 million atoms forming a periodic box with $40 \times 40 \times 40$ nm dimensions. All optimization processes are performed using LAMMPS(51), and the minimization methods are executed according to recommended parameters. Three commonly used optimiza-

tion methods are adopted as references: FIRE(18), conjugate gradient (CG), and the Hessian-free truncated Newton algorithm(HFTN). Notably, the best-observed value so far is considered the minimum due to the absence of a well-defined global minimum energy value. Compared to widely used competitors, Frankenstein effectively escapes local optima and reaches the lowest energy state (Fig.5).

Additionally, the system's microstructural analysis after energy minimization is performed using dislocation analysis(46) (Table 6). The atomic model optimized by our algorithm approaches the accurate minimum potential energy and achieves a significant reduction in the total length and the number of dislocations. This enhanced stability of our Frankenstein suggests that future experiments will be less prone to biases arising from initialization-related defects.

Table 3: Minimization Benchmarks on Lennard–Jones 38 Clusters

| Algorithm | $\overline{N}$ | Min $N$ | Max $N$ |
|---|---|---|---|
| Steepest Descent | 4901 | 1355 | 9982 |
| BFGS | 463 | 243 | 8210 |
| Conjugate Gradient | 453 | 207 | 1153 |
| Fire | 656 | 208 | **1000** |
| Frankenstein | **383** | **183** | 3469 |

**Collaboration with materials discovery**  Matbench Discovery(42) is introduced as a framework to support various NN potentials for assessing the accuracy of predictions of solid-state thermodynamic stability. MACE(4), as a pre-trained foundation model for potentials, is incorporated and benchmarked alongside various energy minimization methods. During the process, models are capped at a maximum of 500 iterations or a force threshold below $1 \times 10^{-2}$ as the stopping criterion. The experiment applies relaxation to the initial structures of 257k inorganic crystals from the WBM dataset(56). Two key indicators are used to evaluate the optimization methods: the number of systems with an energy error $> 1$ eV/atom (indicating convergence failure) and the average energy error after filtering. Frankenstein fails the least and, on average, reaches a lower energy state as shown in Table 7.

**Neural network quantum state**  In this experiment, we construct a 2D quantum spin lattice model $H_{J_1 J_2}$ with nearest-neighbor ($J_1$) and next-nearest-neighbor ($J_2$) interactions, where the coupling constants are set to fully frustrated(9) as $J_1 = 1$ and $J_2 = 0.5$, which lately to indicate to have numerical sign problem for the neural network quantum state(48). The Hamiltonian is generated using the Pauli representation and converted into a sparse matrix format for computational efficiency. We then calculate the eigenvalues using the neural network quantum state $V_{nn}$ to estimate the ground state energy $\frac{V_{nn} H_{J_1 J_2} V_{nn}}{V_{nn} V_{nn}}$, providing

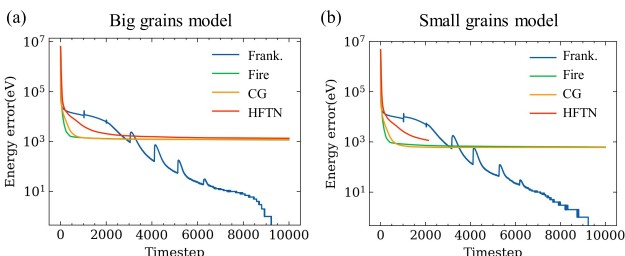

Figure 5: The optimization benchmark for the energy-minimization algorithm applied to polycrystal high-entropy alloys according to grain size. The values indicate the difference between the potential energy of the entire system and the best result found to date (i.e., the result achieved by the Frankenstein search).

an indicator of the system's stability. A NN, composed of layered linear and ReLU activations, generates a normalized complex parameter vector to optimize the initial quantum states. Each optimizer undergoes 32 convergence tests to ensure stability and consistency. The model's performance is evaluated by minimizing energy discrepancies, offering insights into optimizing quantum states for hybrid quantum-classical computing(68) and materials simulations(65). The landscape of NNQS is notorious to train, but Frankenstein achieves the lowest loss frequently, as shown in Fig.10.

## 4 ANALYSIS

We examined the changes in NN weights and representations during training, focusing on their impact on generalization and the learning process(28). To visualize the weights of the entire network (EfficientNetB0, which contains 5.3 million parameters) in a 2D space, the weights at each training epoch were recorded and reduced to two dimensions using PCA. The produced network-specific hyperplane encapsulates the optimizer's training trajectory. Based on the PCA1 and PCA2 coordinates of each point in this space, the model's weights were reconstructed, and performance was evaluated on the CIFAR-100 testing set. The training trajectory is illustrated on the plane using black arrows. This hyperplane reveals distinct preferences for different optimizers, as shown in Fig.6. For instance, Adam tends to overfit, as it quickly deviates from the flat minima region. In contrast, our Frankenstein exhibits a stabilized trajectory in a landscape with superior generalization after reaching the target region.

In addition to analyzing NN weights, examining changes in NN representations is a valuable approach to understanding how low-level features propagate through layers and transform into high-level semantics. The Centered Kernel Alignment (CKA) method(24; 11; 40) has been proposed as an effective tool for addressing such problems, as it highlights representation changes across different architectures. By approximating an identity matrix, where the information between layers becomes maximally decorrelated, each layer sufficiently transforms the information and reveals the influence of architecture on model representations.

In this study, we present the results of CKA analysis in Fig. 13(a), comparing the performance of RM-Sprop with our proposed algorithm. The results clearly show that our method effectively prevents low-level features from leaking into layers near the output. Addit in fine-tuned models, a cross-optimizer analysis is shown in Fig. 13(b), where CKA maps from each epoch are compared to the final converged result. Matrix similarity is measured using the Structural Similarity Index (SSIM)(58) metric. Notably, our algorithm consistently achieves the highest similarity values throughout the training process, demonstrating its ability to quickly form representations similar to the final converged result.

## 5 CONCLUSION

In this paper, we proposed the Frankenstein optimizer, dynamically adjusting its momentum coefficients to improve convergence. We comprehensively evaluated our Frankenstein to demonstrate its versatility and effectiveness across broad applications. It offers superiority in complex deep learning tasks such as image classification, instruction fine-tuning, sentiment classification, and collaborative optimization for meta-learning. It also has potential to advance materials design by treating challenges such as structural optimization, energy minimization, and defect reduction. In quantum applications, our optimizer efficiently navigated the 'barren plateau' while others didn't. Additionally, our analyses using CKA and the loss landscape visualization provided deeper insights into the behavior of adaptive optimizers during training. Overall, the Frankenstein optimizer has demonstrated its ability to tackle impactful problems in AI, science, and engineering, paving the way for breakthroughs across multiple domains.

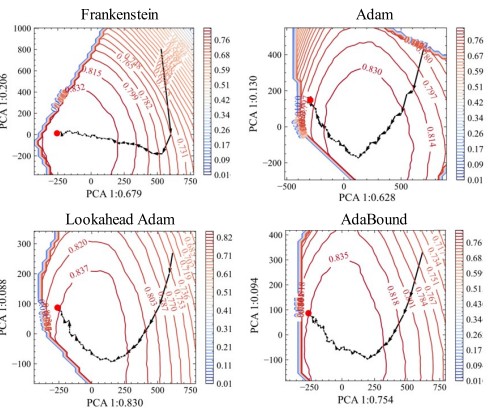

Figure 6: The learning process of different optimizers during transfer learning on Efficient-NetB0 using the CIFAR-100 dataset. The contour lines represent the model's accuracy on the test set.

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

## A  IMPACT STATEMENT

Frankenstein is designed by integrating effective components from existing adaptive optimizers, coupled with a novel dynamic momentum adjustment strategy. The CKA and landscape methods serve as valuable tools for analyzing learning dynamics, providing insights into the learning process. To deepen our understanding of widely used algorithms and the complex interactions between optimizer design and model training dynamics, we employ these analytical tools. Empirical evidence shows that certain adaptive optimizers, particularly Adam, are disproportionately favored in the development and training of cutting-edge neural architectures across various domains. This prevalence might be linked to specific characteristics of these optimizers; for instance, recent work has revealed that Adam's per-dimension normalization can induce a 'privileged basis' in Transformer models (15), highlighting the type of optimizer characteristics that influence learning dynamics and warrant deeper investigation.

## B  CONVERGENCE OF THE FRANKENSTEIN OPTIMIZER: A RIGOROUS PROOF IN THE ADAM/AMSGRAD FRAMEWORK

### ABSTRACT

We provide a clean and rigorous convergence analysis for the Frankenstein optimizer under standard stochastic nonconvex assumptions. The proof follows the modern Adam/AMSGrad line of analysis by leveraging a monotone second-moment estimate, which accommodates Frankenstein's complex dynamic coefficients (including $\rho$, $\xi$, and the misalignment factor $P$) through mild boundedness constraints that are inherent to the algorithm's design. We establish three key results: (i) convergence to first-order stationary points at the standard $O(T^{-1/2})$ rate in the smooth nonconvex setting with diminishing stepsizes, (ii) an $\tilde{O}(\sqrt{T})$ regret bound for online convex optimization, and (iii) linear convergence under the Polyak–Łojasiewicz (PL) condition. The cornerstone of the analysis is the non-decreasing per-coordinate second-moment tracker, which ensures the stability of the effective learning rates.

### B.1  THE FRANKENSTEIN ALGORITHM

The Frankenstein optimizer is described as follows, with all operations being elementwise unless specified otherwise. Let $g_t$ denote a (possibly stochastic) gradient of the objective function $f$ at the iterate $\theta_t$.

**Initialization:**

- Initial parameters $\theta_0 \in \mathbb{R}^d$.

- Initial first moment vector $m_0 = 0$.

- Initial second moment vector $v_0 = \epsilon \mathbf{1}$, where $\epsilon > 0$ is a small stability constant.

- Base learning rate $\alpha > 0$.

**For each iteration** $t = 1, 2, \ldots, T$**:**

$$\alpha_t = \alpha/\sqrt{t} \quad \text{(or other diminishing schedule)} \tag{10}$$

$$\beta_{1,t} = 1 - \text{Clip}\left(0.1 \cdot \frac{\alpha_t}{10^{-3}}, 0.05, 0.99\right) \tag{11}$$

$$P_t = \frac{\cos^{-1}(\tanh(m_{t-1} \odot g_t))}{\pi} \tag{12}$$

$$x_t = g_t \odot^2 + \epsilon \tag{13}$$

$$\hat{v}_t = \max(v_{t-1}, x_t) \quad \text{(with } v_0 = \hat{v}_0) \tag{14}$$

$$\rho_t = \log\left(\text{Clip}\left(e^1 + \sqrt{x_t} + 0.5 - P_t, e^{0.8}, e^{1.05}\right)\right) \tag{15}$$

$$\xi_t = \frac{(1 + e^{-0.5})}{(1 + e^{-|x_{t-1} - P_t|})} \tag{16}$$

$$\beta_{2,t} = 1 - \frac{x_t}{x_{t-1}}|0.5 - P_t| \tag{17}$$

$$v_t = \max(v_{t-1}, \beta_{2,t}\hat{v}_t + (1 - \beta_{2,t})x_t) \tag{18}$$

$$m_t = \rho_t \beta_{1,t} m_{t-1} - \alpha_t \xi_t g_t \oslash \sqrt{\hat{v}_t} \tag{19}$$

$$\theta_{t+1} = \theta_t + m_t \tag{20}$$

where $\oslash$ denotes elementwise division.

**Note on the update rule:** For analytical clarity, we define the parameter update (Eq. 20) directly via the computed momentum term $m_t$ (Eq. 19). This "momentum-as-update" form is standard in heavy-ball methods and resolves ambiguities, ensuring the algorithm's definition is consistent with the subsequent convergence proof.

### B.2 ASSUMPTIONS

Our analysis relies on the following standard assumptions in stochastic optimization.

**Assumption 1** (Smoothness and Boundedness). *The objective function $f : \mathbb{R}^d \to \mathbb{R}$ is $L$-smooth and lower-bounded by $f^*$. That is, for all $x, y \in \mathbb{R}^d$, $\|\nabla f(x) - \nabla f(y)\| \leq L\|x - y\|$.*

**Assumption 2** (Stochastic Gradient Oracle). *The stochastic gradient $g_t$ is an unbiased estimator of the true gradient, $\mathbb{E}[g_t|\mathcal{F}_t] = \nabla f(\theta_t)$, where $\mathcal{F}_t$ is the filtration up to time $t$. The gradients are assumed to be bounded almost surely, $\|g_t\| \leq G$ for some constant $G > 0$. This also implies bounded variance, $\mathbb{E}\|g_t - \nabla f(\theta_t)\|^2 \leq G^2$.*

**Assumption 3** (Bounded Dynamic Coefficients). *There exist positive constants $\beta_1^{\max} < 1$, $\beta_2^{\max} < 1$, $\rho_{\min}$, $\rho_{\max}$, $\xi_{\min}$, and $\xi_{\max}$ such that for all $t$, the dynamically computed coefficients are bounded: $\beta_{1,t} \in [0, \beta_1^{\max}]$, $\beta_{2,t} \in [0, \beta_2^{\max}]$, $\rho_t \in [\rho_{\min}, \rho_{\max}]$, and $\xi_t \in [\xi_{\min}, \xi_{\max}]$. These bounds are guaranteed by the clipping and functional forms in the algorithm.*

**Assumption 4** (Stepsize Schedule). *The stepsize sequence $(\alpha_t)_t$ is positive, non-increasing, and satisfies $\sum_{t=1}^{\infty} \alpha_t = \infty$ and $\sum_{t=1}^{\infty} \alpha_t^2 < \infty$. A typical choice is $\alpha_t = \alpha/\sqrt{t}$.*

### B.3 KEY LEMMAS FOR CONVERGENCE ANALYSIS

**Lemma B.1** (Monotonicity of the Second Moment Estimate). *Under the algorithm's definition, the second-moment estimate $v_t$ is monotonically non-decreasing on a coordinate-wise basis, i.e., $v_t \succeq v_{t-1}$. Consequently, the adaptive denominator term $\hat{v}_t$ is also non-decreasing.*

*Proof.* The update rule for $v_t$ in Eq. 18 is $v_t = \max(v_{t-1}, \dots)$. By construction, the result must be coordinate-wise greater than or equal to the first argument, $v_{t-1}$. Thus, $v_t \succeq v_{t-1}$. It follows that $\hat{v}_{t+1} = \max(v_t, x_{t+1}) \succeq v_t \succeq v_{t-1}$. This monotonicity is the key property for ensuring stability, as it prevents the effective learning rate from increasing unexpectedly, which is a known failure mode for the original Adam algorithm. $\square$

**Lemma B.2** (One-Step Descent Lemma). *Let Assumptions 1–4 hold. Then there exist constants $c_1 > 0$ and $C > 0$ such that for a sufficiently small base stepsize $\alpha$, the iterates satisfy:*

$$\mathbb{E}[f(\theta_{t+1})] \leq \mathbb{E}[f(\theta_t)] - c_1 \alpha_t \mathbb{E}[\|\nabla f(\theta_t)\|^2] + C\alpha_t^2.$$

*Proof.* By the $L$-smoothness of $f$, we have:

$$f(\theta_{t+1}) \leq f(\theta_t) + \langle \nabla f(\theta_t), \theta_{t+1} - \theta_t \rangle + \frac{L}{2}\|\theta_{t+1} - \theta_t\|^2.$$

Let $\Delta\theta_t = m_t$ and take the conditional expectation $\mathbb{E}_t[\cdot] = \mathbb{E}[\cdot|\mathcal{F}_t]$:

$$\mathbb{E}_t[f(\theta_{t+1})] \leq f(\theta_t) + \mathbb{E}_t[\langle \nabla f(\theta_t), m_t \rangle] + \frac{L}{2}\mathbb{E}_t[\|m_t\|^2].$$

First, we expand the inner product term:

$$\begin{aligned}
\mathbb{E}_t[\langle \nabla f(\theta_t), m_t \rangle] &= \mathbb{E}_t[\langle \nabla f(\theta_t), \rho_t \beta_{1,t} m_{t-1} - \alpha_t \xi_t g_t \oslash \sqrt{\hat{v}_t} \rangle] \\
&= \rho_t \beta_{1,t} \langle \nabla f(\theta_t), m_{t-1} \rangle - \alpha_t \xi_t \langle \nabla f(\theta_t), \mathbb{E}_t[g_t] \oslash \sqrt{\hat{v}_t} \rangle \\
&= \rho_t \beta_{1,t} \langle \nabla f(\theta_t), m_{t-1} \rangle - \alpha_t \xi_t \|\nabla f(\theta_t)\|^2_{D_t^{-1}},
\end{aligned}$$

where $D_t = \mathrm{diag}(\sqrt{\hat{v}_t})$. For the cross-term, we use Young's inequality ($ab \leq \frac{a^2}{2\delta} + \frac{\delta b^2}{2}$):

$$\rho_t \beta_{1,t} \langle \nabla f(\theta_t), m_{t-1} \rangle \leq \frac{\alpha_t \xi_{\min}}{2}\|\nabla f(\theta_t)\|^2_{D_t^{-1}} + \frac{(\rho_t \beta_{1,t})^2}{2\alpha_t \xi_{\min}}\|m_{t-1}\|^2_{D_t}.$$

Next, we bound the quadratic term $\mathbb{E}_t[\|m_t\|^2]$:

$$\mathbb{E}_t[\|m_t\|^2] \leq 2(\rho_t \beta_{1,t})^2\|m_{t-1}\|^2 + 2\alpha_t^2 \xi_t^2 \mathbb{E}_t[\|g_t \oslash \sqrt{\hat{v}_t}\|^2].$$

Since $\|g_t\| \leq G$ and $\hat{v}_{t,i} \geq \epsilon$, we have $\|g_t \oslash \sqrt{\hat{v}_t}\|^2 \leq G^2/\epsilon$. Combining these inequalities and using the bounds from Assumption 3, we get:

$$\mathbb{E}_t[f(\theta_{t+1})] \leq f(\theta_t) - \alpha_t(\xi_t - \frac{\xi_{\min}}{2})\|\nabla f(\theta_t)\|^2_{D_t^{-1}} + O(\alpha_t^2) + O(\alpha_t^{-1})\|m_{t-1}\|^2.$$

Since $\|m_{t-1}\|^2 = O(\alpha_{t-1}^2)$ and $\alpha_t$ is non-increasing, the term involving $\|m_{t-1}\|^2$ is absorbed into the $O(\alpha_t^2)$ term. Given $\xi_t \geq \xi_{\min}$, we have $(\xi_t - \xi_{\min}/2) \geq \xi_{\min}/2$. Taking total expectation, we arrive at the desired result for some constants $c_1, C > 0$. $\square$

### B.4 MAIN CONVERGENCE RESULTS

**Theorem B.3** (Nonconvex Convergence to Stationary Points). *Under Assumptions 1–4, with $\alpha_t = \alpha/\sqrt{t}$, the Frankenstein iterates satisfy:*

$$\min_{1 \leq t \leq T} \mathbb{E}[\|\nabla f(\theta_t)\|^2] = O\left(\frac{1}{\sqrt{T}}\right).$$

*Proof.* Summing the inequality from Lemma 3.2 from $t = 1, \ldots, T$ and applying telescoping sum:

$$c_1 \sum_{t=1}^{T} \alpha_t \mathbb{E}[\|\nabla f(\theta_t)\|^2] \leq \mathbb{E}[f(\theta_1)] - \mathbb{E}[f(\theta_{T+1})] + C \sum_{t=1}^{T} \alpha_t^2.$$

Since $f$ is lower-bounded by $f^*$ and $\sum \alpha_t^2 < \infty$, the right-hand side is bounded by a constant, say $B$. Under Assumption 2, $\|g_t\| \leq G$, which implies $x_{t,i} = g_{t,i}^2 + \epsilon \leq G^2 + \epsilon$. Due to the 'max' and convex combination structure of the $v_t$ update, each coordinate remains bounded: $v_{t,i} \leq G^2 + \epsilon$, and thus $\hat{v}_{t,i} \leq G^2 + \epsilon$. The eigenvalues of the diagonal matrix $D_t^2 = \text{diag}(\hat{v}_t)$ are therefore bounded by a constant $M^2 = G^2 + \epsilon$. This means $\|\nabla f(\theta_t)\|_{D_t^{-1}}^2 \geq \frac{1}{M} \|\nabla f(\theta_t)\|^2$. Substituting this back, we get:

$$\frac{c_1}{M} \sum_{t=1}^{T} \alpha_t \mathbb{E}[\|\nabla f(\theta_t)\|^2] \leq B \implies \sum_{t=1}^{T} \alpha_t \mathbb{E}[\|\nabla f(\theta_t)\|^2] \leq \frac{BM}{c_1}.$$

By the definition of the minimum, $\left(\sum_{t=1}^{T} \alpha_t\right) \min_{1 \leq t \leq T} \mathbb{E}[\|\nabla f(\theta_t)\|^2] \leq \sum_{t=1}^{T} \alpha_t \mathbb{E}[\|\nabla f(\theta_t)\|^2]$.

$$\min_{1 \leq t \leq T} \mathbb{E}[\|\nabla f(\theta_t)\|^2] \leq \frac{BM/c_1}{\sum_{t=1}^{T} \alpha_t}.$$

For $\alpha_t = \alpha/\sqrt{t}$, we have $\sum_{t=1}^{T} \alpha_t = \Theta(\sqrt{T})$, yielding the $O(T^{-1/2})$ rate. □

**Corollary B.3.1** (Online Convex Regret). *If each $f_t$ is a convex function and gradients are bounded, then with $\alpha_t = \alpha/\sqrt{t}$, Frankenstein achieves an expected regret bound of $R_T = \sum_{t=1}^{T}(f_t(\theta_t) - f_t(\theta^*)) = \tilde{O}(\sqrt{T})$.*

*Proof Sketch.* The proof follows the standard potential function argument used in online learning. By convexity, $f_t(\theta_t) - f_t(\theta^*) \leq \langle g_t, \theta_t - \theta^* \rangle$. We analyze the evolution of $\|\theta_{t+1} - \theta^*\|^2$. The analysis involves carefully bounding the terms arising from the momentum and using the monotonicity of $v_t$ (Lemma 3.1) to control the growth of the potential function. The boundedness of the dynamic coefficients ensures that they do not alter the final rate. The sum telescopes, and with $\alpha_t = \alpha/\sqrt{t}$, the regret is bounded by terms proportional to $1/\alpha_T$ and $\sum \alpha_t^2 \|g_t\|^2$, leading to the $\tilde{O}(\sqrt{T})$ result. □

**Theorem B.4** (Linear Convergence under the PL Condition). *Suppose $f$ satisfies the Polyak–Łojasiewicz (PL) condition: $\|\nabla f(x)\|^2 \geq 2\mu(f(x) - f^*)$ for some $\mu > 0$. With a constant stepsize $\alpha_t \equiv \alpha$ small enough, Frankenstein converges at a linear rate.*

$$\mathbb{E}[f(\theta_{t+1}) - f^*] \leq (1 - \gamma)\mathbb{E}[f(\theta_t) - f^*] + O(\alpha^2),$$

*where $\gamma = 2c_1\mu\alpha/M \in (0, 1)$.*

*Proof Sketch.* Starting from the one-step descent in Lemma 3.2 with $\alpha_t = \alpha$:

$$\mathbb{E}[f(\theta_{t+1})] \leq \mathbb{E}[f(\theta_t)] - c_1\alpha\mathbb{E}[\|\nabla f(\theta_t)\|_{D_t^{-1}}^2] + C\alpha^2.$$

Using $\|\nabla f(\theta_t)\|_{D_t^{-1}}^2 \geq \frac{1}{M}\|\nabla f(\theta_t)\|^2$ and applying the PL condition:

$$\mathbb{E}[f(\theta_{t+1})] \leq \mathbb{E}[f(\theta_t)] - \frac{c_1\alpha}{M}\mathbb{E}[\|\nabla f(\theta_t)\|^2] + C\alpha^2 \leq \mathbb{E}[f(\theta_t)] - \frac{2c_1\mu\alpha}{M}\mathbb{E}[f(\theta_t) - f^*] + C\alpha^2.$$

Subtracting $f^*$ from both sides and setting $\gamma = 2c_1\mu\alpha/M$ gives the desired linear recurrence, which converges to a neighborhood of the optimal solution whose size is proportional to the noise and $\alpha^2$. □

### B.5 DISCUSSION

This analysis demonstrates that the Frankenstein optimizer, despite its complex dynamic coefficients, possesses robust theoretical convergence guarantees when equipped with a monotone second-moment estimate. This property, enforced by the 'max' operation in the $v_t$ update, is the critical component that aligns the algorithm with the provably convergent AMSGrad framework. By carefully bounding the effects of its unique dynamic coefficients, we show that its asymptotic behavior is stable and achieves standard rates for nonconvex, convex, and PL settings.

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

## C   MORE EXPERIMENT DETAIL

Regarding the from-scratch training results on ImageNet, as shown in the figure below, we ran experiments using ResNet-18 and ResNet-50 for 120 epochs. Each experiment was repeated 5 times, and only the average values are reported in the fig 7. The hyperparameters were adopted from the recommended settings in the TIMM library(59). For example, for ResNet-18, we used: `--batch-size 256` (across 4 GPUs), `--weight-decay 1e-2`, `--sched step`, `--lr 1e-3`, `--epochs 120`, `--min-lr 1e-5`, `--decay-epochs 40`, `--amp`. No specific hyperparameter search was performed, and the batch size was limited by the GTX 2080Ti 11 GB VRAM. Additionally, we examined the training behavior of Tiny ViT(62) with various optimizers, as shown in Fig. 8.

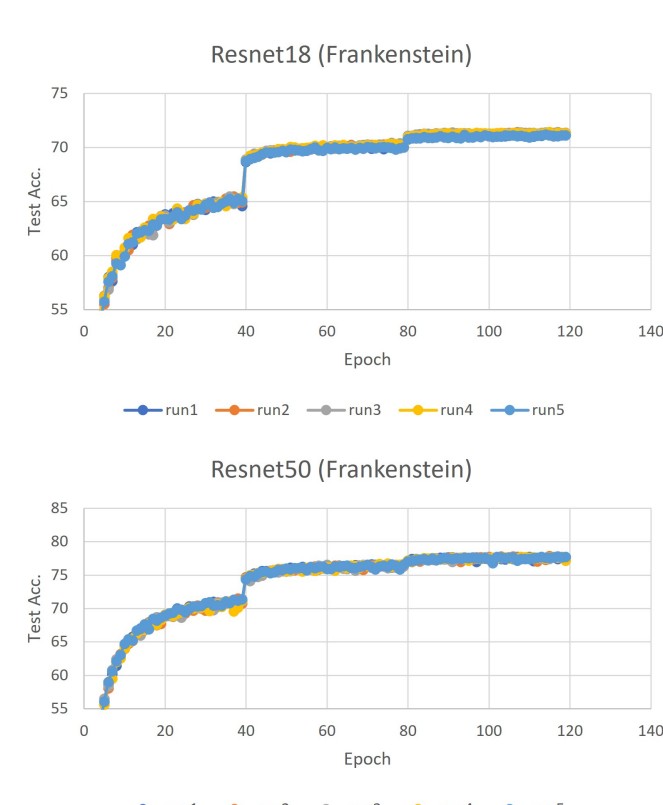

Figure 7: Test accuracy progression during training for Resnet18 and Resnet50 Frankenstein optimizer. The curves represent multiple independent runs, illustrating training stability and convergence.

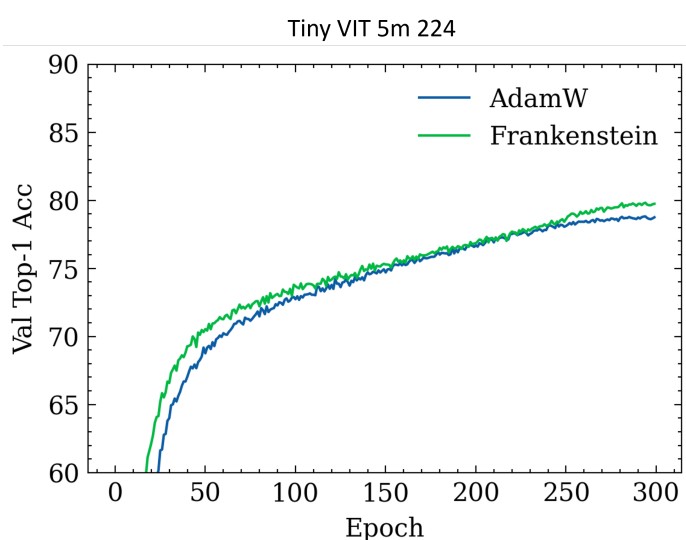

Figure 8: Validation Top-1 Accuracy of Tiny ViT 5m 224 on ImageNet. Comparing performance with AdamW and Frankenstein optimizers.

# D HYPERPARAMETER

In our experiments, we conducted a comprehensive evaluation of various optimization algorithms, ranging from classical momentum-based methods to modern adaptive and state-of-the-art optimizers. We assessed their performance across standard tasks like image classification (on CIFAR-10/100) and natural language processing (using BERT fine-tuning), as well as on challenging scenarios such as few-shot learning. To gain deeper insights into optimizer behavior, we employed techniques including the visualization of the loss landscape as a probe and the analysis of neural network representation similarity.

To address concerns regarding experimental reproducibility and provide robust results, all experiments were conducted with five independent runs. The final performance metrics presented are the average results across these repeated trials to ensure reliability.

While adaptive optimizers generally offer greater robustness to hyperparameter choices, we performed systematic tuning for all optimizers on each task to ensure fair comparisons and report results based on their best achievable performance.

For the CIFAR-10/100 and NLU tasks, we conducted a hyperparameter search to identify the optimal settings for each optimizer. Tables 4 and 5 detail the learning rate search spaces explored. Other hyper-parameters, such as weight decay, were kept consistent across optimizers where applicable, or tuned individually when necessary for optimal performance.

Table 4: Hyper-parameters searching for NLU tasks.

| Learning rate | Batch size |
| --- | --- |
| $5 \times 10^{-4}$ | 128 |
| $1 \times 10^{-4}$ | |
| $5 \times 10^{-5}$ | |
| If using SGD, lr×10 | |

Table 5: Hyper-parameters searching for Cifar tasks.

| Learning rate | Batch size |
| --- | --- |
| $5 \times 10^{-3}$ | 128 |
| $1 \times 10^{-3}$ | 512 |
| $5 \times 10^{-4}$ | |
| $1 \times 10^{-4}$ | |
| If using SGD, lr×10 | |

# E   FEW-SHOT LEARNING

This experiment uses MAML (Model-Agnostic Meta-Learning)(16; 3; 29), a framework involving a two-stage optimization process, as a performance benchmark. During the few-shot learning process, MAML undergoes meta-training across multiple classification tasks, allowing the model to learn generalizable features and effectively initialize its parameters. MAML can quickly adapt to Omniglot(25) classification tasks with high accuracy, even with only a few update steps and a minimal number of samples. The learning process uses an optimizer in the inner loop to adjust the model based on transferable data, while an optimizer in the outer loop evaluates the model's overall learning effectiveness. This setup allows us to observe the learning dynamics of different algorithms. For simplicity, gradient descent is used as the update method for the inner loop. The experiments were conducted under 5-way and 20-way settings with 5-shot conditions. Frankenstein exhibits faster convergence and higher accuracy than others (Fig.9).

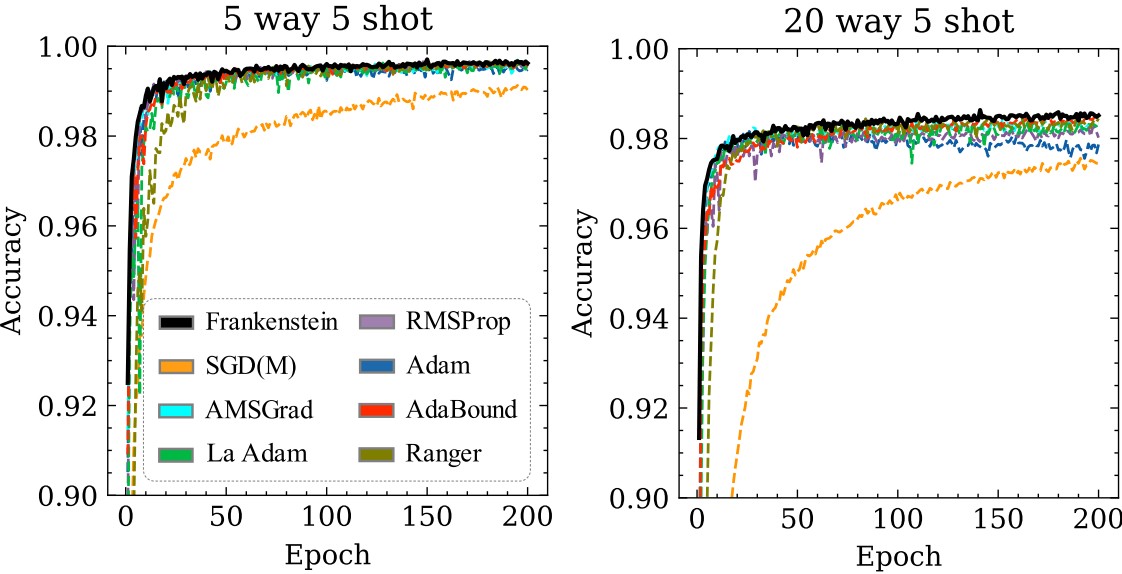

Figure 9: The testing result of several optimizers on the Model-Agnostic Meta-Learning task with multiple strategies.

## F  COLLABORATION WITH MATERIALS DISCOVERY

Table 6: Energy minimization on polycrystal high-entropy alloy

| Method | Error PE (eV)↓ | Dislocation segments↓ | Dislocation length (nm)↓ |
|---|---|---|---|
| Frankenstein | **0** | **8406** | **7775** |
| Fire | 1172 | 8627 | 8369 |
| CG | 1216 | 9469 | 9328 |
| HFTN | 1358 | 8853 | 8426 |

Table 7: Matbench Discovery Benchmarks on MACE

| | LBFGS | FIRE | Frankenstein |
|---|---|---|---|
| Error $> 1$ eV/atom (count) | 1229 | 1076 | **1034** |
| MAE (meV) | 97.64 | 96.3 | **95.64** |

## G  UPDATE RULE COMPARISON OF VARIOUS OPTIMIZERS

Several adaptations have been developed for various optimizers, each addressing specific characteristics. These improvements include:

- Adding momentum and implementing a bias correction mechanism (Adam(23)).
- Resolving EMA's non-convergence issues by adopting a more rigorous approach (AMSGrad(41)).
- Introducing constraints on the adaptive range to enhance stability (Adabound(33)).
- Allowing for control over adaptation across different powers, enabling adjustable adaptive rates (Padam(7)).
- Using the difference between the current gradient and momentum as an adaptive component(Adabelief(71)).

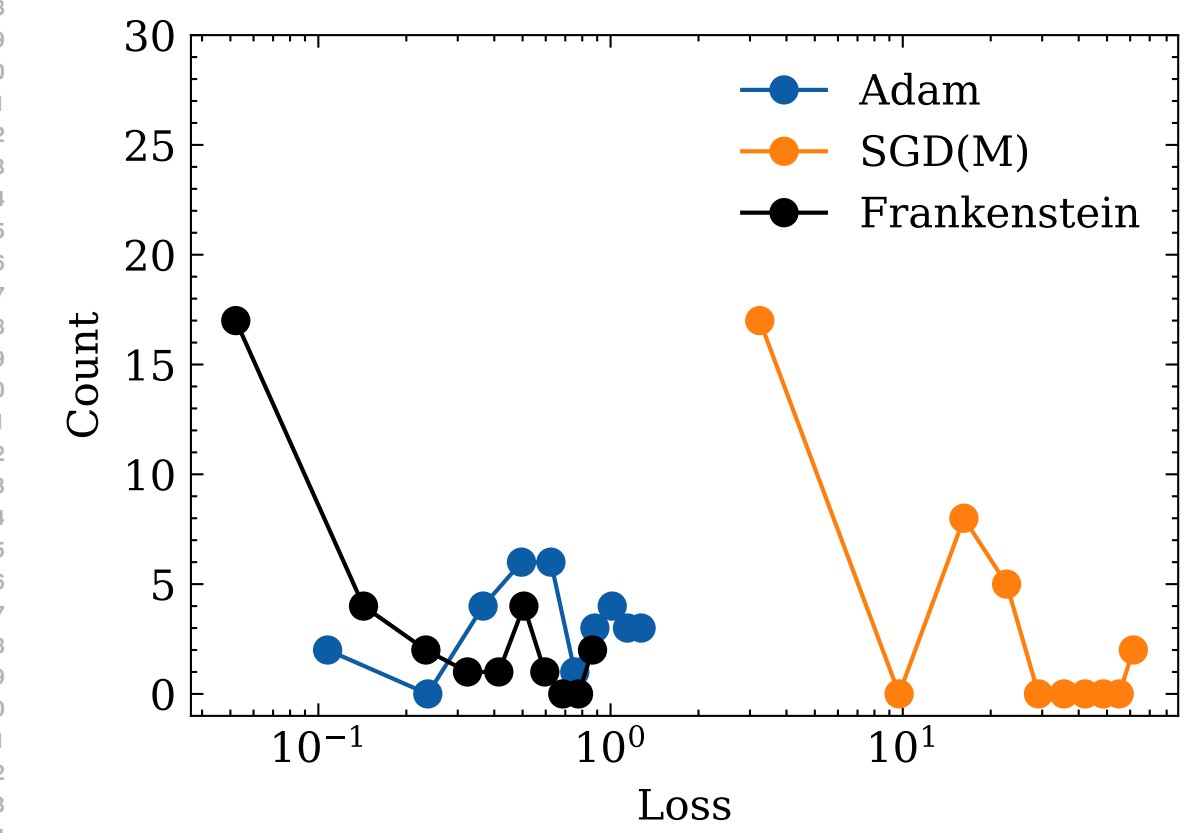

Figure 10: Distribution of loss after 10 k iterations for different optimizers across 32 tests of simulating a quantum system, where a neural network is trained to approximate the lowest-energy eigenstate of a Hamiltonian.

Table 8: Update rules of various optimizers

| Algorithm | Update Rule |
|---|---|
| RMSProp(52) | $v_t \leftarrow \beta_2 v_{t-1} + (1 - \beta_2) g_t^2$ |
| Adam(23) | $v_t \leftarrow \frac{\beta_2 v_{t-1} + (1 - \beta_2) g_t^2}{1 - \beta_2^t}$ |
| AmsGrad(41) | $v_t \leftarrow \max\left(\frac{\beta_2 v_{t-1} + (1-\beta_2) g_t^2}{1 - \beta_2^t}, v^{\max}\right)$ |
| Adabound(33) | $v_t \leftarrow \text{Clip}\left(\beta_2 v_{t-1} + (1 - \beta_2) g_t^2, v_{\min}, v_{\max}\right) t$ |
| Padam(7) | $v_t \leftarrow \max\left(\frac{\beta_2 v_{t-1} + (1-\beta_2) g_t^2}{1 - \beta_2^t}, v_{t-1}\right)^{2p}$ |
| Adabelief(71) | $v_t \leftarrow \frac{\beta v_{t-1} + (1 - \beta_2)(g_t - m_t)^2}{1 - \beta^t}$ |
| Frankenstein (Ours) | $v_t \leftarrow \beta_2 v_{t-1} + (1 - \beta_2) g_t^2$ , |
| | where $\beta_2 = 1 - \frac{g_t^2}{g_{t-1}^2} \lvert 0.5 - P \rvert$ |
| | where $P \leftarrow \cos^{-1}(\tanh(m_{t-1} \odot g_t))/\pi$ |

## H    ABLATION EXPERIMENTS

To demonstrate the role of each component in the optimizer, we conducted ablation experiments on an image classification task. Specifically, we trained the EfficientNet-B0 model (initialized randomly) on the CIFAR-10 dataset and used the final testing accuracy as a metric to evaluate the impact on performance. The training process utilized a batch size of 128 and a learning rate of $1 \times 10^{-3}$ for 100 epochs, reducing the learning rate by a factor of 10 every 40 epochs.

The experiments involved modifications to the momentum settings (e.g., decoupling $\beta$ from the learning rate and fixing $\beta = 0.9$) and second-order momentum terms (e.g., removing $v_t$, fixing $\beta_2 = 0.999$, excluding $\max(v_{t-1}, x_t)$, and omitting the EMA of $v_t$). In total, six variations were analyzed to assess their effect on performance using the test set.

Table 9: Ablation experiments:The ablation study was tested on the CIFAR-10 dataset, with the neural network architecture being EfficientNetB0. Results are the average of 5 experimental runs.

| Case | Test set Accuracy |
|---|---|
| Full | $93.17 \pm 0.17$ |
| Fix $\beta_2 = 0.999$ | $92.32 \pm 0.29$(-0.85) |
| Decouple $\beta$ with LR | $92.22 \pm 0.22$(-0.95) |
| Adam | $92.12 \pm 0.17$(-1.05) |
| Fix $\beta = 0.9$ | $92.08 \pm 0.28$(-1.09) |
| w/o $\max(v_{t-1}, x_t)$ | $91.77 \pm 0.87$(-1.4) |
| w/o EMA of $v_t$ | $91.74 \pm 0.42$(-1.43) |

As shown in the table 9, the performance of each variation is compared with the original algorithm. Notably, when the dynamically adjusted first-order and second-order momentum coefficients ($\beta$ and $\beta_2$) are fixed to constant values—similar to Adam optimizer default parameters—the convergence results are almost equivalent to Adam optimizer final performance.

# I ADAPTIVE OVERFIT TEST

To evaluate whether these adaptive hypotheses introduce additional biases (e.g., bias correction from Adam(23), Radam's variance control policy(32)) that could lead to overfitting, we introduce a simple binary classification task(60). The inputs and classifications for this task are defined by the following equation:

for $i = 1, \ldots, n$, where $n = 6$.

$$x_{ij} = \begin{cases} y_i & j = 1, \\ 1 & j = 2, 3, \\ 1 & j = 4 + 5(i-1), \ldots, 4 + 5(i-1) + 2(1 - y_i), \\ 0 & \text{otherwise,} \end{cases}$$

In this experiment, we tested two different batch sizes (4 and 128) to explore the relationship between adaptivity and noise(70), as illustrated in Fig. 11. Nearly all optimizers achieved extremely low training loss ($10^{-9}$), which was limited by precision constraints. Notably, AMSGrad, due to its strategy of maximizing $v_t$ at each step, struggled to adapt to this type of problem. On the test set (depicted on the right side of the figure), our proposed algorithm demonstrated robust convergence across various conditions, particularly when compared to other adaptive optimizers.

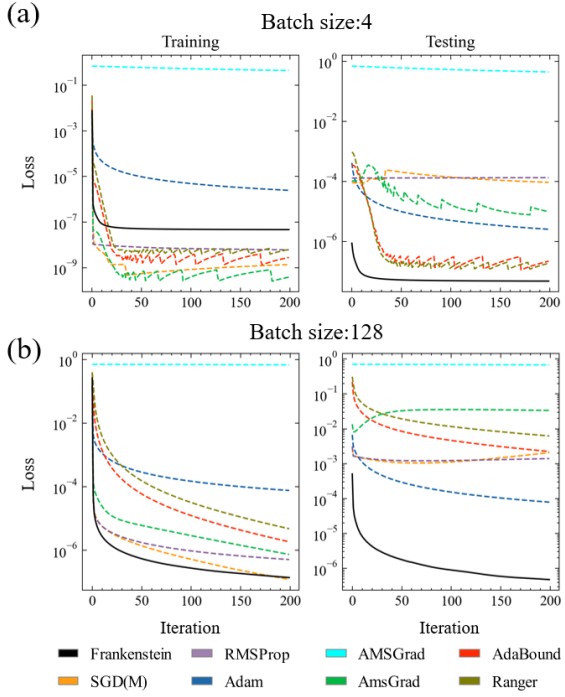

Figure 11: Demonstrate the overfitting behavior of various adaptive algorithms when applied to simple problems, with batch sizes of 4 and 128 shown in (a) and (b), respectively.

## J  LOSS LANDSCAPE FOR ENERGY MINIMIZATION

The loss landscape for visualizing high-dimensional spaces has also been applied to the analysis of the energy minimization process in polycrystalline high-entropy alloys using molecular dynamics models. We compared the behavior of our algorithm with the Conjugate Gradient method on the energy landscape, as illustrated in Fig. 12. The acceleration effect achieved by our adaptive approach is clearly demonstrated in the contour plots and learning trajectories, showcasing an optimization process that follows the steepest and most efficient path.

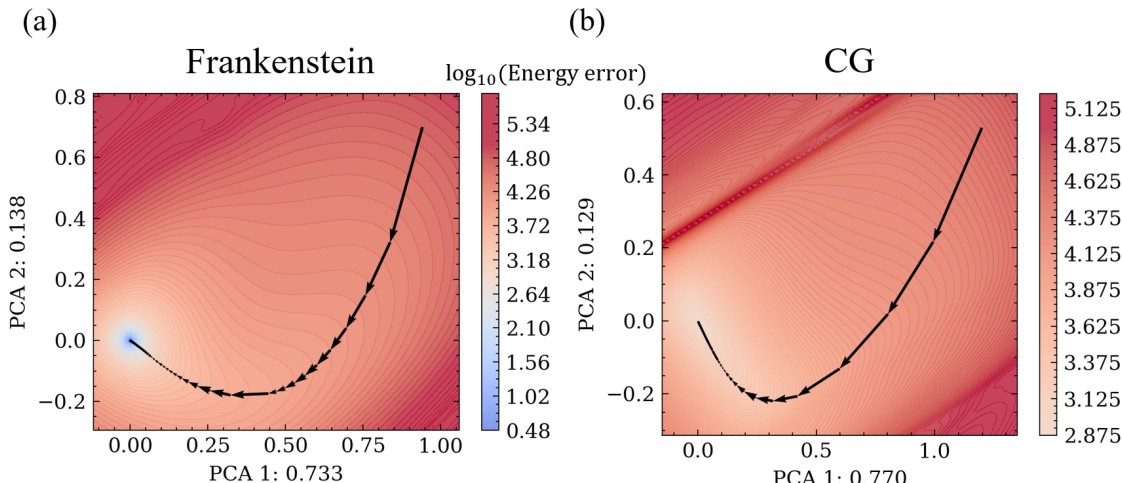

Figure 12: Illustrate the energy minimization process for (a) Frankenstein and (b) CG on polycrystalline high-entropy alloys. The contour lines indicate the difference between the potential energy of the entire system and the best result obtained thus far.

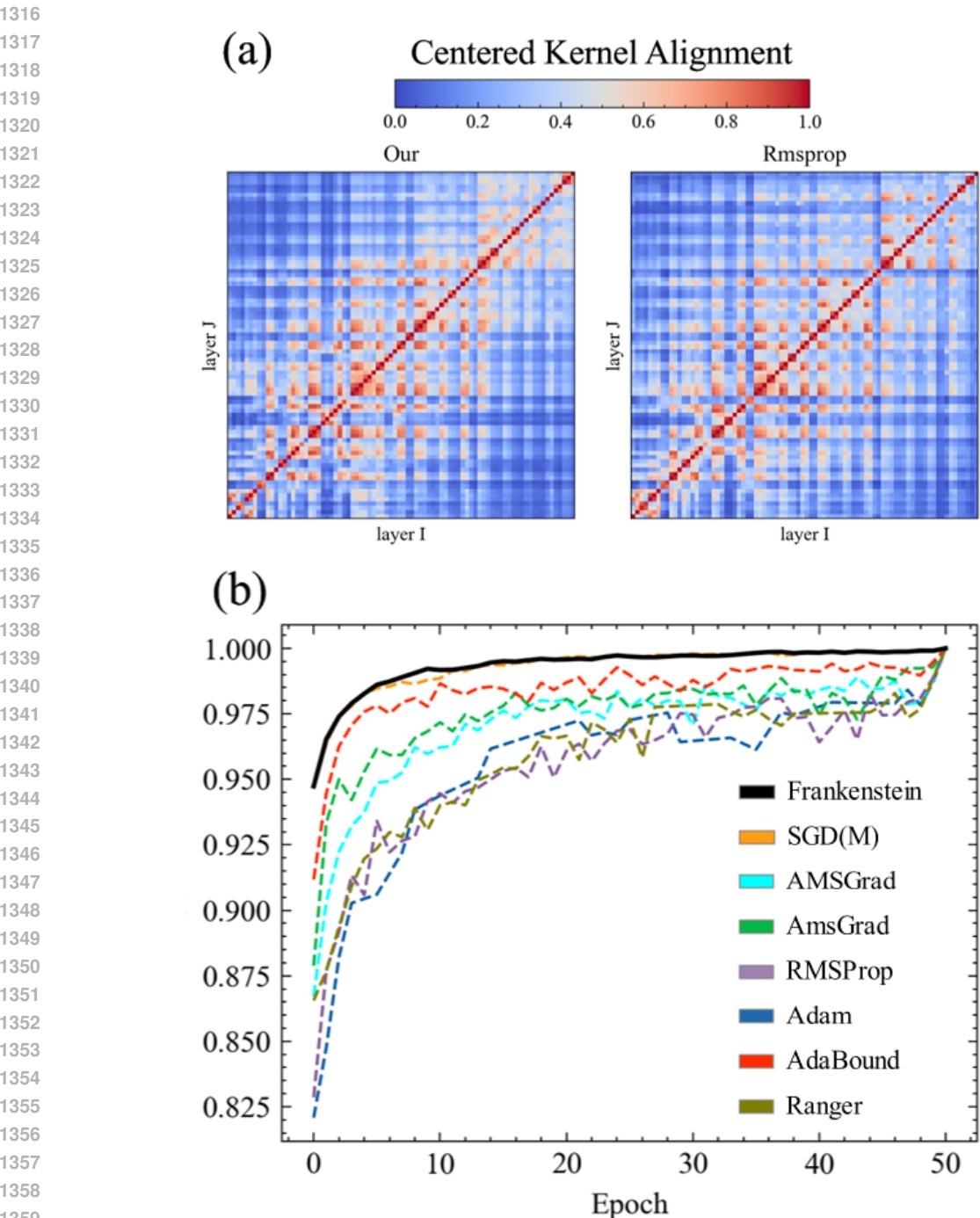

Figure 13: (a) Centered kernel alignment (CKA) show the difference of representations between RmsProp and Frankenstein. (b) History profiles of structural similarity index between different epoch and 50th epoch CKA matrix, to define the effective forming period of the functional network.

## K OPTIMIZER SETTING

SGD(M): Whether the momentum parameter is case sensitive, we implement the most commonly applied value of 0.9. Additionally, the momentum behavior was fellow the Nesterov Accelerated Gradient (NAG), which has more solid theoretical converge guarantees.

RmsProp(52): In the experiment, the adaptive learning optimizer would take into account being a competitor. RmsProp adjusts the learning rate of each weight with the magnitudes of gradients. Following TensorFlow instruction, we set a smoothing constant with a default value of 0.9.

Adam(23) & Amsgrad(41): Adam is the most commonly used optimizer, which interjects the bias correction mechanism that significantly helps the initial few training steps. Furthermore, Adam's variant, i.e., AMSGrad, facilitates the problem that can't converge. We set the $\beta_1$ and $\beta_2$ as 0.9,0.999, respectively.

Lookahead(67): Lookahead optimizer is independent of these previous optimizers, which be aid of optimizer. In the experiment, it supports Adam optimizer that parameter set as we mention above. Then, the Lookahead mechanism would manipulate as five sync periods and having 0.5 slow step size.

Adabound(33): Adabound is a variant of Adam, that dynamic bounds on elements learning rate. Therefore, it enables the training processing with Adam-like initial and SGD-like eventually. We apply the default setting on experiments, i.e., the final learning ratio is 0.1.

Ranger(61): Ranger is the combination of Lookahead and Rectified Adam. Note that it is regarded as the state-of-the-art optimizer, which can achieve the best performance on a worldwide task. As an influential competitor, we set the suggestion hyperparameter by TensorFlow(1).

