# OpenReview forum: "Frankenstein Optimizer: Harnessing the Potential by Revisiting Optimization Tricks"
_ICLR.cc/2026/Conference — ICLR 2026 Conference Withdrawn Submission_

### Official Review · Reviewer_1gfg · 2025-10-30

**Soundness:** 2
**Presentation:** 2
**Contribution:** 2
**Rating:** 4
**Confidence:** 4

**Summary:**

This paper introduces the Frankenstein Optimizer, a unified adaptive optimization algorithm that dynamically adjusts both first- and second-momentum coefficients based on gradient alignment, scale ratio, and an acceleration factor. The method integrates insights from Adam, AMSGrad, and AdaBelief to maintain consistent learning dynamics and react quickly to sudden gradient changes. The authors provide theoretical convergence guarantees and evaluate the optimizer across vision, NLP, and scientific simulation tasks, showing faster convergence and improved generalization compared to existing adaptive methods.

**Strengths:**

1. The paper presents a well-motivated and unified adaptive optimization framework that dynamically adjusts both first- and second-momentum coefficients based on training state variables ($P$, $\rho$, $\xi$).

2. It offers clear theoretical grounding with convergence analysis consistent with AMSGrad, and provides extensive experiments covering computer vision, NLP, and scientific simulation tasks.

3. Ablation studies are thorough, demonstrating the contribution of each component.
The inclusion of representation similarity (CKA) and loss landscape visualization adds interpretability and provides valuable insights into the optimizer’s generalization behavior.

**Weaknesses:**

1. The proposed method is conceptually sound but remains largely incremental. While the idea of dynamically adjusting both first- and second-momentum coefficients is reasonable, it closely overlaps with recent adaptive optimizers such as Adan, Sophia, Lion, and AdaBelief. The presented formulation appears to integrate and refine these existing strategies rather than introducing a fundamentally new optimization principle.

2. The experiments span a broad range of domains, including computer vision, NLP, and physical simulation, which demonstrates the method’s general applicability. However, the analysis lacks depth: several key ablation and sensitivity studies are missing, making it difficult to isolate the contribution of each proposed component (e.g., $P$, $\rho$, $\xi$). In particular, it remains unclear whether the observed improvements arise from the dynamic momentum design itself or from tuning advantages.

3. The paper allows $\beta_{2,t}$ to take negative values under certain conditions, but it provides insufficient numerical validation to demonstrate that the algorithm remains stable in such cases. This design choice raises concerns about potential instability, especially in mixed-precision training or when gradients fluctuate sharply. The theoretical analysis does not explicitly address this behavior, and no experiments are presented to verify robustness under these challenging scenarios.

4. Figures and tables have inconsistent font sizes and aspect ratios, and some plots are blurry or poorly scaled. The appendix contains multiple blank part, resulting in a cluttered layout that negatively affects readability and professionalism.

**Questions:**

1. It would be helpful to clarify more concretely how Frankenstein differs from recent adaptive optimizers such as Adan, Sophia, and Lion. In particular, what makes its optimization principle or dynamic coefficient design genuinely distinct?

2. Since $\beta_{2,t}$ can become negative in some cases, how do the authors ensure the method remains numerically stable? Please mention whether any clipping or safeguard mechanisms are applied, and consider including evidence under extreme gradient conditions.

3. The algorithm introduces additional dynamic terms ($P$, $\rho$, $\xi$). Do these components noticeably increase computational cost or training time compared to Adam? Even a brief runtime or FLOPs comparison would help readers understand the efficiency trade-off.

4. The overall presentation could be improved, especially in figure layout and formatting. Unifying font sizes and scaling in the final version would make the paper look much cleaner and easier to read.

---

### Official Review · Reviewer_MXSh · 2025-10-30

**Soundness:** 2
**Presentation:** 1
**Contribution:** 1
**Rating:** 2
**Confidence:** 5

**Summary:**

This paper introduces the Frankenstein optimizer, a new adaptive optimizer that dynamically adjusting its momentum coefficients. While conventional SGD-like and Adam-like optimizers tend to use a fixed momentum coefficients, Frankenstein dynamically adjusts its first and second momentum coefficients and introduces a new acceleration factor. Empirical results across diverse tasks show the outstanding performance of Frankenstein optimizer in both convergence speed and test accuracy.

**Strengths:**

Both convergence analysis and experimental evaluation are conducted for the Frankenstein optimizer, showing its advantage over other optimizers.

**Weaknesses:**

1. The paper is very difficult to read and understand. The description of the algorithm in Section 2 is dense and the notation is not always clear. (Please refer to Q1, Q4 as examples).

2. Some claims in this paper lack evidence. No experiments and no citations. Even if some claims are trivial to the authors, there should be clear evidence for them, like citations at least. Otherwise, the paper is unconvincing. (Please refer to Q2, Q3 as examples)

3. Just like 2, the Frankenstein optimizer itself is also lacking sufficient evidence for its insights. This optimizer needs a quite complex update procedure, but the ideas are mostly heuristic. Although the experiments demonstrate the outperformance of Frankenstein, the paper fails to explain why, and to how much the mechanisms within the optimizer are essential. It will be much convincing if the authors can conduct sufficient ablation studies to validate the effectiveness of the all the mechanisms within this optimizer.

4. Frankenstein optimizer requires a subtle hyperparameter tuning, but there are no hyperparameter sensitivity analysis. Moreover, it is unclear how the baselines are tuned. If the baseline optimizers are not tuned as extensively as Frankenstein, the comparison is unfair.

**Questions:**

1. In Section 2, the paper suddenly introduces the Equation 1-3 without any background information, citation and motivation claim. What is the Verlet integration? What is $\Delta x$? Why is it important to the optimization in deep learning?

2. In Line 102, the paper claims "If the training process enters a stable phase where the gradient remains relatively constant and the momentum closely matches the gradient, the momentum coefficient should be maintained as specified in Equation 2; otherwise, this balance could be disrupted." Why? What is Equation 2 for? What is the mentioned "balance"?

3. In Line 109, the paper claims "To maintain training consistency, the decay rate caused by momentum and the momentum coefficient should correspond to the gradient and the learning rate; otherwise, using a learning rate decay strategy might lead to overly rapid convergence, requiring additional iterations to achieve the optimal convergence point." Is this claim contradictory? Why the overly rapid convergence require additional iterations to converge? Besides this problem, what is your evidence to this claim?

4. In Line 131, the paper says that "$ \rho ← log(e^1 + \sqrt{v_t} + 0.5 − P )$ where $x_t$ refers the summation of squared gradient $g_t^2$ and a small constant $\epsilon$." Where is $x_t$?

5. Talking about the coefficient scheduling in SGD and Adam, there are some existing practices [1-3]. What is the advantage of Frankenstein against them?

6. Minor: The citation style in this paper is uncommon, compared to other ICLR papers. Why are you using such a style?

**Reference**

[1] Yanli Liu, Yuan Gao, and Wotao Yin. An Improved Analysis of Stochastic Gradient Descent with Momentum. Advances in Neural Information Processing Systems, 2020.

[2] John Chen, Cameron Wolfe, Zhao Li, Anastasios Kyrillidis. Demon: Improved Neural Network Training with Momentum Decay. IEEE International Conference on Acoustics, Speech and Signal Processing (ICASSP), 2022.

[3] Xianliang Li, Jun Luo, Zhiwei Zheng, Hanxiao Wang, Li Luo, Lingkun Wen, Linlong Wu, Sheng Xu. On the Performance Analysis of Momentum Method: A Frequency Domain Perspective. International Conference on Learning Representations (ICLR), 2025.

---

### Official Review · Reviewer_F7gE · 2025-11-01

**Soundness:** 3
**Presentation:** 2
**Contribution:** 2
**Rating:** 4
**Confidence:** 4

**Summary:**

This paper introduces the Frankenstein optimizer, an adaptive optimization algorithm that aims to combine the fast convergence of adaptive methods with the strong generalization of SGD. The proposed method dynamically adjusts both first- and second-order moment coefficients ($\beta_1$ and $\beta_2$) at each training step. These adjustments are based on the training state, including the learning rate, the alignment between the current gradient and past momentum, and the volatility of squared gradients. Eexperiments are conducted on computer vision, NLP, few-shot learning, and scientific simulation tasks, to demonstrate the method's superior performance in both convergence speed and final accuracy.

**Strengths:**

1. The use of dynamic $\beta_1$ and $\beta_2$ is intuitively well-motivated.
2. Frankenstein has shown strong empirical results in both convergence speed and final accuracy/performance across multiple test domains.

**Weaknesses:**

1. The proposed algorithm is overly complex, combining numerous "tricks" into a single framework. The design feels highly heuristic and lacks principled motivation for its specific functional forms. For example, the exact functional forms for $\rho$ (the log function) and the acceleration factor $\xi$ (the sigmoid-like structure) are not rigorously derived. The paper would be stronger if it provided more justification for these specific choices over other potential ones. In addition, the algorithm contains multiple hardcoded constants (e.g., 0.1, 10e-3, 0.5, $e^{0.5}$, $e^{1.05}$). The paper provides no information on how these values were selected or how sensitive the optimizer's performance is to them.
2. While the breadth of experiments is a strength, several choices and presentation details raise concerns about the evaluation. (1) The presentation in Figure 2 is confusing. The caption states it shows results on "(a) CIFAR-10 and (b) CIFAR-100," but the plots themselves are titled "(a) Training set" and "(b) Testing set." This makes it unclear which dataset is being used for which plot. (2) In Figure 2, the authors present results from fine-tuning pre-trained EfficientNet models (B0-B6) on CIFAR-10/100. However, the ablation study in Table 9 is conducted by training an EfficientNetB0 from scratch on CIFAR-10. Results from these two different training paradigms (fine-tuning vs. from-scratch) are not directly comparable, making it difficult to connect the main results to the ablation analysis. (3) The language modeling experiment in Figure 4 is limited to LoRA fine-tuning of LLaMA-7B. While this is a relevant task, optimizers can behave very differently during from-scratch pre-training. A more convincing demonstration would involve training a smaller language model from scratch to assess the optimizer's stability and effectiveness in a pre-training scenario.
3. The ablation study presented in Table 9 is insufficient to validate the complex design of the Frankenstein optimizer. The study is performed on a small-scale task (EfficientNetB0 on CIFAR-10), which may not be representative of the optimizer's behavior on more complex tasks like ImageNet or LLM training. Furthermore, the ablation only removes entire components (e.g., "w/o EMA of vt"). It fails to analyze the most questionable aspects of the design: the specific functional forms and the hardcoded constants. An ablation that compares the chosen form of $\rho$ or $\xi$ against simpler alternatives, or that shows the effect of varying the constants, is essential for justifying the algorithm's complexity.

**Questions:**

See above.

---

### Official Review · Reviewer_cwu7 · 2025-11-08

**Soundness:** 3
**Presentation:** 3
**Contribution:** 2
**Rating:** 4
**Confidence:** 3

**Summary:**

This paper presents Frankenstein, a new optimizer that tries to combine the fast convergence of Adam with the good generalization of SGD. It uses three main ideas. First, an adaptive first-momentum term linked to the learning rate. Second, a dynamic second-moment term that uses a max-tracking approach similar to AMSGrad. Third, an acceleration factor based on a misalignment measure called P. The authors test it across a wide range of tasks including image classification, NLP, few-shot learning, and scientific simulations. They claim it performs better or on par with strong baselines and provide theoretical analysis and interpretive tools such as CKA and loss landscape visualizations.

**Strengths:**

* Addresses a known challenge in optimization by trying to get both speed and generalization.
* Very broad experiments across vision, language, meta-learning, and physics-based tasks.
* Shows consistent improvements or comparable performance with strong baselines like Adam, Sophia, Adan, and SGD.
* Includes theoretical convergence analysis and interpretive experiments.
* Contains ablations and analysis to explain how different components contribute to performance.

**Weaknesses:**

* The algorithm is inconsistent. The main text, appendix, and tables show slightly different versions. The convergence proof applies only to one version that does not match the main algorithm. It is unclear which version was used in the experiments. This makes the results hard to reproduce.
* The optimizer is very complex. It combines multiple dynamic components including P, rho, xi, beta1, and beta2. The reasoning behind the exact formulas is not clear. It looks empirical and heuristic. This makes it hard to implement correctly and understand.
* There is no real analysis of computational cost. Faster convergence is measured only in steps or epochs. With all the extra calculations per step, the optimizer might actually be slower in wall-clock time than Adam or SGD.
* Some experiments are not fully fair. Baselines are sometimes trained for different numbers of epochs, some architectures are modified, and hyperparameter details are sparse.
* Hyperparameter sensitivity is not explored. The optimizer has many internal constants and clipping ranges, but we do not know how performance changes if these are adjusted.
* Writing and presentation could be clearer. Some formulas are scattered and variable definitions are inconsistent, which makes the text hard to follow.

**Questions:**

1. Which exact version of Frankenstein was used in the experiments? Please provide one clear pseudocode.
2. How much extra compute does Frankenstein require in wall-clock time, memory, or FLOPs?
3. How sensitive is the optimizer to its internal constants and clipping ranges?
4. What exactly is the misalignment factor P? Is it per parameter, per layer, or a single scalar?
5. How does the dynamic beta2 term behave? Does the negative beta2 combined with the max really maintain monotonicity?
6. Can the convergence proof be adapted to match the actual implementation? Right now it assumes scalar smooth updates, but the algorithm is coordinate-wise with clipping and max.
7. Will the authors release code, seeds, and full configurations for reproducibility?

---

### Note · Authors · 2026-01-23

**Comment:**

We have read and agree with the venue’s withdrawal policy, and we hereby withdraw this submission (Submission #24241).
After carefully considering the reviews, we believe the current version requires substantial revisions, especially regarding the clarity and consistency of the method description, experimental rigor, and the discussion of efficiency/stability.
We sincerely thank the reviewers for their constructive feedback, and we will incorporate these suggestions in a revised and more mature version for a future submission.

**Withdrawal Confirmation:**

I have read and agree with the venue's withdrawal policy on behalf of myself and my co-authors.